# Elucidation of transient protein-protein interactions within carrier protein-dependent biosynthesis

Thomas G. Bartholow[1], Terra Sztain[1], Ashay Patel[1], D. John Lee[1,3], Megan A. Young [1], Ruben Abagyan[2] & Michael D. Burkart [1✉]

Fatty acid biosynthesis (FAB) is an essential and highly conserved metabolic pathway. In bacteria, this process is mediated by an elaborate network of protein•protein interactions (PPIs) involving a small, dynamic acyl carrier protein that interacts with dozens of other partner proteins (PPs). These PPIs have remained poorly characterized due to their dynamic and transient nature. Using a combination of solution-phase NMR spectroscopy and protein-protein docking simulations, we report a comprehensive residue-by-residue comparison of the PPIs formed during FAB in *Escherichia coli*. This technique describes and compares the molecular basis of six discrete binding events responsible for *E. coli* FAB and offers insights into a method to characterize these events and those in related carrier protein-dependent pathways.

[1] Department of Chemistry and Biochemistry, University of California, San Diego, La Jolla, CA, USA. [2] Skaggs School of Pharmacy and Pharmaceutical Sciences, University of California San Diego, La Jolla, USA. [3]Present address: Bioengineering and Therapeutic Sciences, University of California San Francisco, San Francisco, CA, USA. ✉email: mburkart@ucsd.edu

Carrier protein-dependent synthases are responsible for the biosynthesis of a vast array of molecules, from primary metabolites to complex natural products[1,2]. These are generally organized as type I or type II enzymes, with the type I "megasynthases" containing multiple enzymatic domains and carrier proteins housed as large multi-domain proteins[3]. In contrast, type II synthases exist as discrete proteins that must recognize and associate with one another in solution through an organized choreography of metabolic steps (Fig. 1A). In *Escherichia coli* fatty acid biosynthesis (FAB) more than 25 partner proteins (PPs) are known to functionally bind to the acyl carrier protein (AcpP) (Fig. 1B)[4–6], a small, four-helix bundle protein that shuttles intermediates between both fatty acid biosynthetic enzymes and regulatory proteins[7]. AcpP must form specific protein•protein interactions (PPIs) with multiple partners, efficiently chaperoning intermediates through 30–35 discrete enzymatic steps to produce the membrane lipids that maintain homeostasis and facilitate cellular reproduction[8,9]. Simultaneously FAB generates fatty acid intermediates for cofactor biosynthesis and secondary metabolism[10]. This study further illustrates AcpP•PP recognition through unique PPIs with each of the FAB enzyme players (Fig. 1A) while presenting a combinatorial method to characterize these transient interactions useful for both engineering and inhibitor design.

Throughout the iterative FAB cycle, substrates and intermediates are not only tethered to AcpP through a 4′-phosphopantetheine (PPant) thioester linkage[11,12], but they are sequestered within the hydrophobic pocket of the AcpP helices, protecting intermediates from non-specific reactivity[13–15]. Reactions are controlled through this sequestration and presentation of the substrates appended to 4′-phosphopantetheine, a process termed chain flipping[13,16]. The rapid doubling times and relatively narrow distribution of fatty acid products require an efficient, high fidelity FAB[16,17], suggesting that stochastic binding events of AcpP with its binding partners are unlikely. Activity studies and mechanism-based crosslinking experiments have demonstrated that acyl-AcpP binding and enzyme turnover are highly specific (Fig. 2C)[17–20]. A growing body of evidence suggests PPIs play an important role in the mechanism of chain flipping and, therefore, the processivity of these pathways[21–24]. For example, recent studies have demonstrated that engineering enzyme specificity for a non-native AcpP-dependent enzyme can be accomplished by modifying the PPI residues for improved binding[25]. Even single atom changes in the identity of AcpP-bound cargo have been demonstrated to impart perturbations to the structure of acyl-AcpP[26,27].

Here we used ¹H-¹⁵N HSQC-NMR titration studies to collect residue-by-residue information for six de novo FAB partner enzymes to characterize each intrinsic PPI with the *E. coli* AcpP. Experiments were performed to study the interfaces of AcpP with elongating ketosynthases FabB and FabF, reductases FabG and FabI, dehydratase FabA, and thioesterase TesA. These spectroscopic data combined with a combinatorial docking protocol benchmarked with crosslinked structures of AcpP in complex with FabA, FabZ, FabB, and FabF provide atomic-resolution information on which residues of AcpP mediate each step in iterative de novo FAB. This combinatorial method was able to overcome the unique challenges of modular synthases, with substrate identity effecting carrier protein structure and each enzyme forming unique interactions with the carrier. Due to the high sequence homology of AcpP with carrier proteins from other species[21,25] and polyketide synthases[28], this protocol is expected to extend for characterization of ACP•PP interactions for engineering and drug design across multiple systems.

## Results

**NMR Titrations reveal dynamic AcpP interface.** Previous work has established the utility of ¹H-¹⁵N HSQC-NMR titrations in the study of rapid and intricate PPIs[29–31]. In this study, uniformly labeled and perdeuterated ¹⁵N-C8-AcpP (octanoyl-AcpP) (Figs. S6 and S11) was subjected to NMR titration using increasing concentrations of unlabeled PPs to detect the residues on AcpP that experience chemical shift migration (Figs. 2A and S1–3). Once saturated with partner enzyme, the extent of peak migration was quantified using the chemical shift perturbation (CSP) calculations (Figs. 2B S1–3b)[32]. The perturbed regions were then projected onto the amino acid sequence and 3D structure of the protein to identify regions affected by PP binding (Figs. 2C, D and 4B). Furthermore, we utilized the TITAN NMR lineshape analysis program[33] to analyze our spectra and obtain thermodynamic and kinetic parameters (Table S9). Previous work has demonstrated the ability of CSPs to identify critical PPIs in carrier protein-mediated biosynthesis[34–36]. Combining our titration data on FabF, FabI, FabG, and TesA with previous titrations on FabB[34] and FabA[24] allowed us to compare the binding interface on AcpP dictating PP recognition, highlighting distinct AcpP residues involved in the binding of specific classes of FAB enzymes

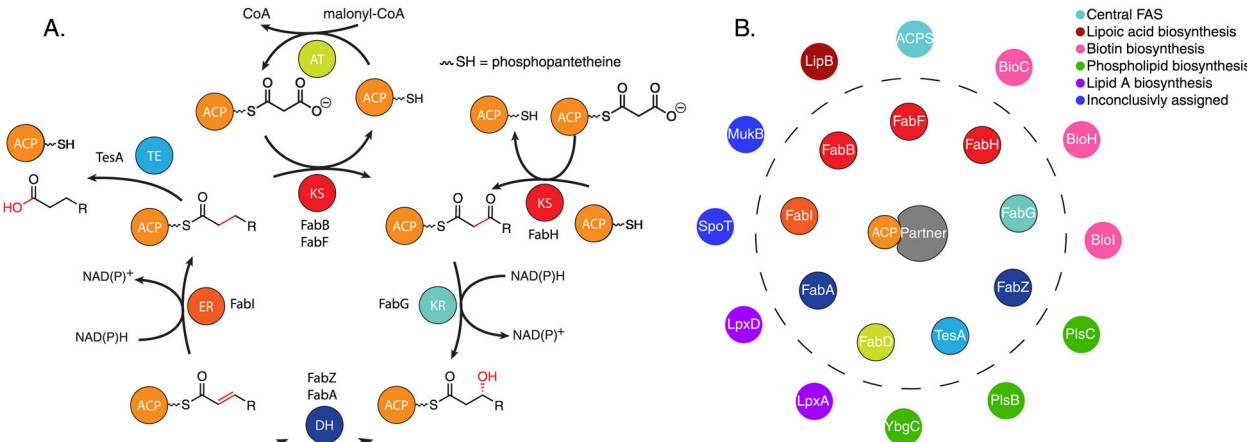

**Fig. 1 The reactions and partners of saturated fatty acid biosynthesis. A** The FAB elongation cycle. ACP acyl carrier protein, KS ketosynthase, KR ketoreductase, DH dehydratase, ER enoylreductase, TE thioesterase, AT acyltransferase. TesA is known to interact with AcpP but is not a participant of *E. coli* FAB. **B** Twenty-nine examples of known AcpP interacting enzymes colored by function; the enzymes within the dotted line are those from FAB, whose color corresponds to the colors of **A**.

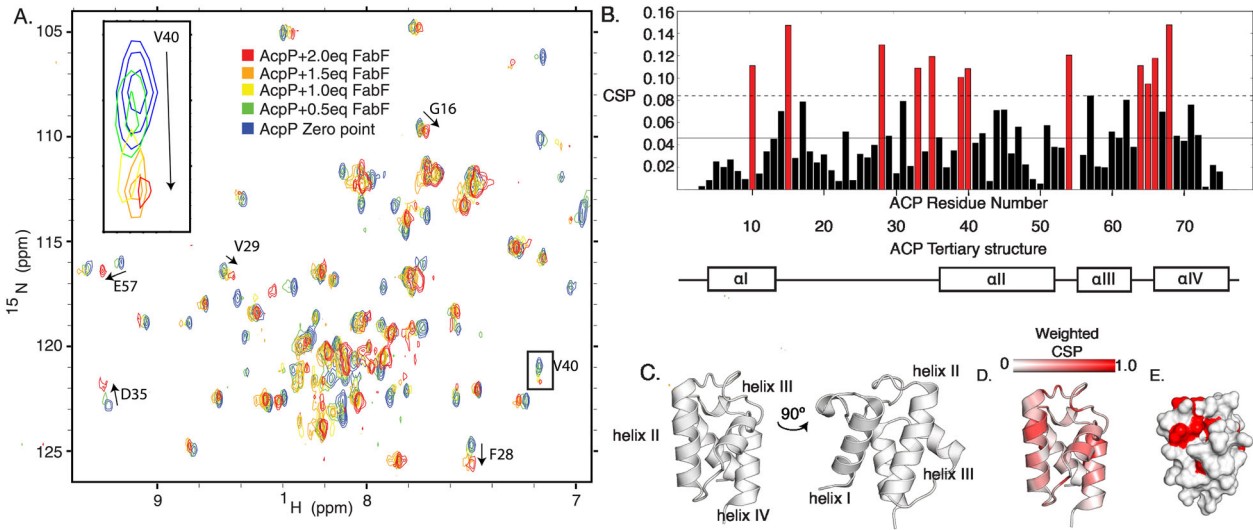

**Fig. 2 ¹H-¹⁵N HSQC titration of C8-AcpP with FabF. A** Five overlaid ¹H-¹⁵N HSQC spectra of ¹⁵N-C8-AcpP titrated with increasing concentrations of unlabeled FabF. The peak migration of V40 is enlarged as an example of a "titration curve." **B** A bar chart of each AcpP residue's CSP with 2.0 molar equivalents of FabF. The mean is shown as a solid line and one standard deviation is the dashed line above. CSPs greater than this value are shown in red.

The CSP equation used was $\mathrm{CSP} = \sqrt{\left(\frac{1}{2}\right)\left[\delta_H^2 + (\alpha \cdot \delta_N)^2\right]}$. An alpha value of 0.2 was used. **C** The tertiary structure of *E. coli* AcpP displaying the classical

interface of helix II and IV, this orientation will be used throughout the paper when displaying the interacting face. This perspective is rotated 90° to display the side face of the ACP. **D** The cartoon structure colored by weighted CSP value, viewed from the interacting face. **E** The surface of the AcpP interacting face with the CSPs one standard deviation above the mean colored in red.

and internal AcpP residues that are likely important to the mechanistic process of chain flipping.

**Docking to elucidate PP-binding site**. The ¹H-¹⁵N HSQC-NMR titrations shown here can provide specific information of the AcpP residues involved in PP binding; however, the residues on the PP that mediate binding cannot be determined by this process. While titrations provide ample evidence of the carrier protein's interacting residues, no information is gained about the residues of the PP with which they are interacting. We have recently elucidated the X-ray crystal structures of several FAB enzymes crosslinked to AcpP, including FabA, FabZ, FabB, and FabF. These structures can indicate residues involved in PPIs on both proteins; however, each structure requires prior development of enzyme-specific crosslinking probes, which are not available in all cases. We sought to develop protein–protein docking protocols with Molsoft's ICM software to predict structures of the AcpP•FabI, AcpP•FabG, and AcpP•TesA complexes that have eluded experimental structural characterization[37,38]. Crystal structures of previously crosslinked AcpP-PPs were used to optimize this protocol, described more fully in the "Methods" section. Briefly, it was identified that to accurately re-create complexes it was necessary to produce a water box in which the PPs were minimized. Docking simulations were carried out between the X-ray structural model of heptanoyl-AcpP-C7 (PDB 2FAD), to which a methylene was added to the acyl chain simulate C8-AcpP, and crystal structural models of binding PPs from which cofactors had been removed. Using expanded calculations to assist the general docking protocol (Fig. 3A), we were able to recapitulate crosslinked structure interfaces (Fig. 3B and Table S1) to sub 7 Å RMSD for the complex and sub 2 Å RMSD at the interface. Crosslinked structures were used in benchmarking as they give a learning set to examine which protocols and docking methods perform well, also demonstrating the ability

of our docking method to recapitulate ACP•PP interfaces. However, it must be noted that the comparison is imperfect, with the docked structures and NMR representing interactions in solution while the crosslinked structures are crystallized and covalently bound in a catalytic conformation. For example, the AcpP structure 2FAD and the AcpP crosslinked to FabA have a ~2 Å RMSD. Furthermore, it should be noted that the crosslinked and *apo* PPs have differing structural similarity: with the FabF structure 1.3 Å RMSD between crosslinked[39] and uncrosslinked[40], FabB 3.5 Å RMSD between crosslinked[34] and uncrosslinked[41], and FabA 4.6 Å RMSD between crosslinked[35] and uncrosslinked[42] structures. The developed protocols were subsequently used to determine the binding interfaces of AcpP•FabI, AcpP•FabG, and AcpP•TesA (Fig. 3C, Table S2, Figs. S4 and S5) in conjunction with the NMR data. This methodology provides valuable context for matching the AcpP interactions to the PP structures, and the breadth of previously reported AcpP and PP activity and mutagenesis studies enable further validation of predicted AcpP•PP against past mutagenic experiments.

**A combinatorial method to characterize modular synthase PPIs**. To judge the ability of a combined NMR and docking method to accurately predict the structures of interacting enzymes, docking was expanded to include CSP information. The interface residues identified through CSPs were given as focus residues for AcpP binding and known interacting residues specified below were given for PPs. Docked models of the enzymes FabA, FabB, and FabF were compared with and without known residues supplied. All the enzymes tested had docked orientations which allowed for chain flipping of the substrate based on the position of Ser 36 and known active sites of the partners, demonstrating the ability of informed calculations to filter out nonproductive complexes. The average RMSD between the docked model and crosslinked crystal structures was 9.29 Å in the

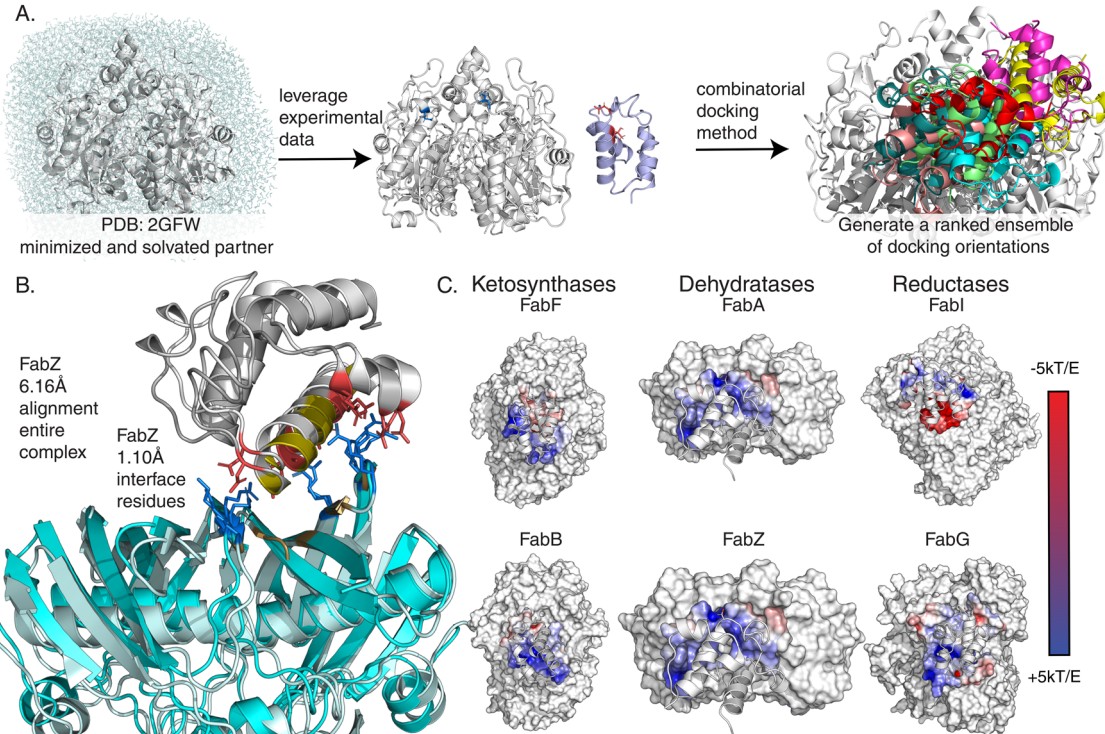

**Fig. 3 Docking workflow and generated models. A** A workflow showing water box generation and minimizations of FabZ (PDB: 6N3P) in purple docked to AcpP (PDB: 2FAD) in gray. The optimized docking protocol is described in more detail in the "Methods" section. Solvation and energy minimization were found to be essential to recapitulate the interface within 3 Å RMSD as shown in **B**. Comparison of the interfaces of the docked model (FabZ: light blue and ACP: dark gray) to the crosslinked crystal structure (FabZ: pale cyan and ACP: light gray). Charged residues within 5 Å of the interfaces are displayed as sticks with the negatively charged (red) and positively charged (blue). Hydrophobic residues are colored gold. **C** The surfaces of six PPs from three families of enzymes with electrostatic potentials of partner enzymes shown within 5 Å of the bound AcpP. Larger versions of these images are presented in Fig. S4.

informed calculation of FabF•AcpP binding and 20.08 Å (Table S7) without experimental knowledge. FabB models were 5.6 Å RMSD in the informed model and 10.5 Å in uninformed, and FabA models were 9.98 Å RMSD in the informed calculation and 9.89 Å in the uninformed. These data demonstrate that while docking alone was able to re-create the crosslinked structure, care must be taken to ensure the models are relevant. Leveraging experimental data ensured greater confidence in the models while yielding a suite of interface structures of relevant interacting orientations. To evaluate the ability of widely available online servers to re-create AcpP•PP interfaces the AcpP•FabF complex was docked using the Cluspro[43–45], HADDOCK[46,47], and Rosie[48,49] servers (Tables S7 and S8). It should be noted that with informed residues and properly prepared structures online servers perform well. The structures used below are taken from the most stable docked orientation for simplicity, though the full suite of orientations remains relevant. However, these methods open the door to more studies of possible preliminary encounter complex states which may facilitate the fully bound form seen in crosslinking.

**Ketosynthases: FabF and FabB.** Elongating ketosynthases iteratively extend acyl-AcpP by two carbon units using malonyl-AcpP as a carbon source via a decarboxylative thia-Claisen condensation (Fig. 1A)[50]. C8-AcpP was titrated with increasing concentrations of the FabF ketosynthase (Fig. 2A and Fig. S9) and compared to recently published data of C8-AcpP titrated with FabB[34]. An octanoyl acylation state was selected to maintain consistency with prior work[34,35] and was utilized for all titrations in this study. Upon titration with FabF, residues beginning at the

end of helix I and the start of helix II of C8-AcpP display significant CSPs. Helix II displays perturbations throughout until there is a small loss at the end of helix II; signals assigned to residues nearly throughout C8-AcpP were perturbed until the end of helix IV where the perturbations drop off. The largest CSPs, residues with CSPs greater than one standard deviation from the mean, in the FabF titration (Table S3) included I10 and L15 on helix 1 and F28 on loop 1 (Fig. 2B); D35, T39, V40 on helix 2; I54 on loop 2; and T64, V65, Q66, and A68 on helix 3. D35 and T39, the charged or polar residues which lie along the interface, appeared within interacting distance of N56′ and Q63′ (residue designators for the PPs in the complexes will be denoted by primes)[34,35,51] (Fig. 4C, D). Surprisingly, a large number of these residues (I10, L15, F28, I54, T64, V65, and A68) are located within the acyl pocket or far from the interface yet show large perturbations. We hypothesized that these interior perturbations represent internal hydrophobic rearrangements that occur upon chain flipping during the binding event. Titan analysis calculated a $K_d$ of $8.3 \pm 9.8\,\mu M$ with a $k_{off}$ of $3512 \pm 3341\,s^{-1}$ and an approximately one to one stoichiometry (Table S9 and Fig. S12).

FabB performs the same ketosynthase reaction as FabF, but performs the first unsaturated elongation step, making a case study in specificity[52]. The NMR titrations were previously performed, but the data will be restated here for comparison. Overall signals from the AcpP•FabB titration display a slightly less broad set of CSPs than those from the AcpP•FabF titration, starting with little perturbation until the top of helix I. Perturbations continue once more at the end of loop I with more sparse interactions on helix II, without perturbation at residues 40 and 41 as well as a drop off in perturbation at the end of helix II. There are two CSPs on loop II and perturbations span

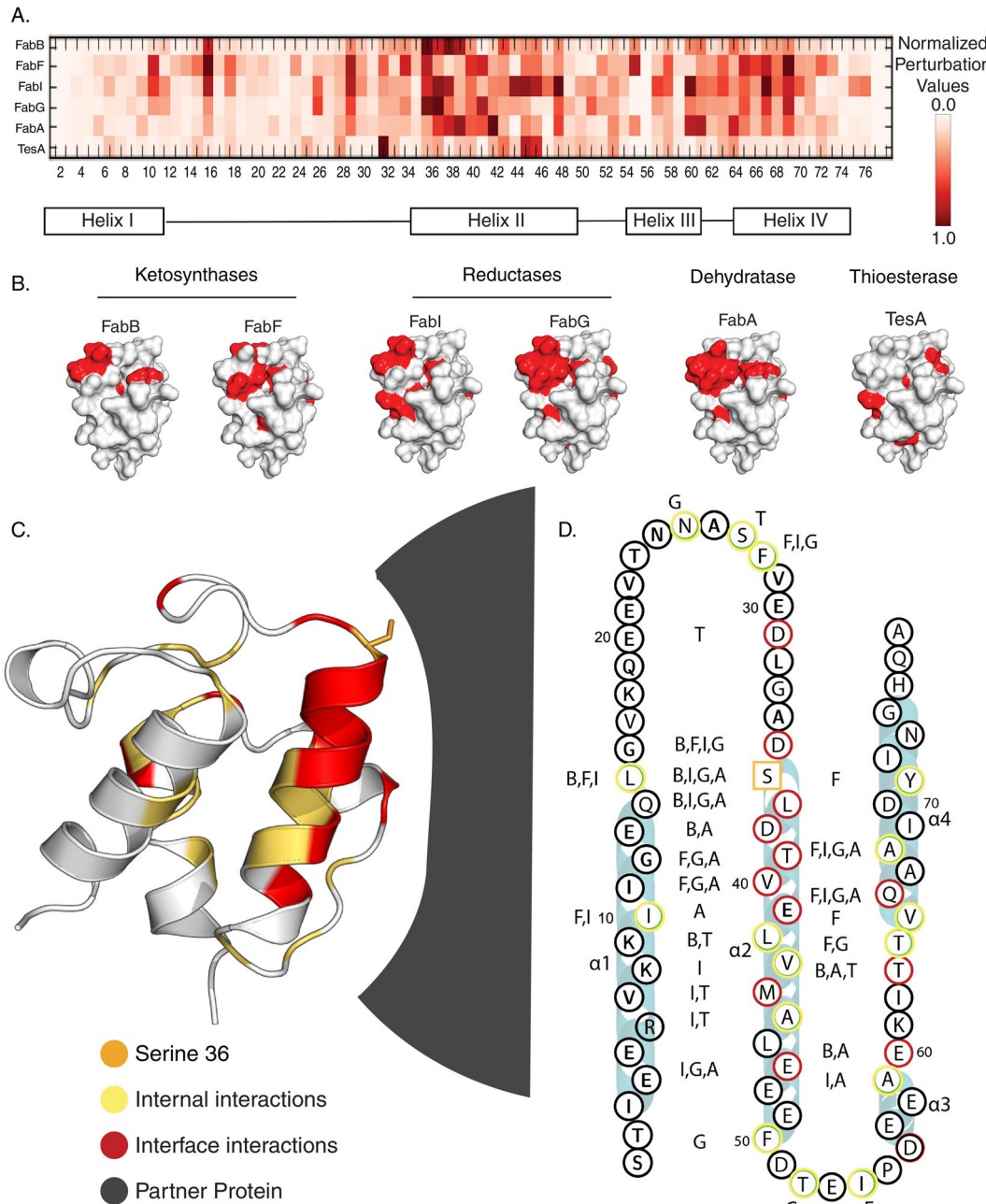

**Fig. 4 Combined data of AcpP interaction with FAB partner enzymes. A** A heat map of CSPs, each titration is normalized and colored in proportion to the largest CSP in the titration. **B** The most perturbed interface of six AcpP partners from the four classes of enzyme, viewed from the interacting face. **C** AcpP (PDB: 2FAD) displayed with the most perturbed residues colored by their interaction, the ACP is rotated 90° from the interacting face. A generic partner surface is shown for context. **D** The C8-AcpP•partner PPIs with the largest CSPs colored based on their interaction with residues displayed in red (interface) or green (internal). S36 is displayed in orange. The PP responsible is labeled alongside the residue. B (FabB), F (FabF), I (FabI), G (FabG), A (FabA), and T (TesA) are used as shorthand.

most of helix III and loop III. Finally, helix IV sees sparse perturbations at the top of helix IV. In FabB the most perturbed residues were L15 of helix 1; D35, S36, L37, D38, and L42 on helix 2; E60 on helix 3; and T63 on loop 3. FabB's effect on AcpP is largest on many acidic residues at the interface, D35, D38, and E60 are all among the most perturbed (Fig. 4). Residues D35 interacts with K62′ and D38 forms a salt bridge with R65′ on FabB. E60 interacts with K150′ in the docked model a small number of hydrophobic residues, such as L15 and L42, exhibit CSPs[34]. Though FabF and FabB both perform the same fundamental chemical reaction, they appear to have distinct

interfaces. FabF has a slightly larger interface (1023 Å$^2$) when comparing docked models with FabB (962 Å$^2$), perhaps consistent with its broader activity and the wider impact on CSPs compared to FabB[53]. This further agrees with data demonstrating a tighter binding for FabF than the previous calculated FabB $K_d$ of $37.6 \pm 6.6\,\mu M$. Though FabB and FabF share particularly similar structures and activity[37,39], their interactions with AcpP are unique.

**Reductases: FabG and FabI**. The condensation reaction performed by ketosynthases generates 3-oxoacyl-AcpP, which is subsequently

reduced to 3R-hydroxyacyl-AcpP by FabG in a NADPH-dependent fashion (Fig. 1A and Fig. S1)[54]. For NMR titrations, $NAD^+$ was added along with FabG, as previous studies showed a difference in AcpP binding efficiency in the presence and absence of $NAD^+$ (Fig. S7)[55]. The total perturbed residues span many residues across the AcpP. The perturbed residues begin at the end of helix I, with a few interactions across loop I. Helix II is perturbed to some degree across most of the AcpP, with only small regions seeing CSPs as low as background. Nearly all residues on loop II and through to helix IV are perturbed until the bottom of helix IV. The residues of C8-AcpP exhibiting the largest CSPs (Table S5 and Fig. S2) were: N25 and F28 on loop 1; D35, S36, L37, T39, V40, E47, and F50 on helix 2; and T64, Q66, and A68 on helix 3. Interface residues D35 and E47 interact with R19′ and R207′ of FabG, respectively, while L37, T38, and V40 form hydrophobic interactions at the interface. N25, F28, F50, T52, T64, Q66, and A68 were all positioned away from the interacting face in the model. The identified region of interaction is in agreement with the binding region previously identified by mutagenesis and activity assays[56,57]. Binding calculations demonstrated a $K_d$ of 52.3 ± 27.5 μM with a $k_{off}$ of 3559 ± 2061 $s^{-1}$ and a two to one stoichiometry (Table S9 and Fig. S13).

The final step of each elongation cycle in (saturated) FAB is catalyzed by the enoylreductase, FabI, which produces a saturated acyl-AcpP through NADH–dependent reduction of enoyl-AcpP (Fig. 1A)[58]. FabI was also titrated with $NAD^+$ present. Upon interaction with FabI, NMR signals from residues throughout C8-AcpP exhibited CSPs (Table S4 and Fig. S8), with perturbations beginning on helix I and showed a few sparse interactions through helix I and onto loop I. However, more interactions are seen on helix II, which shows interactions throughout only diminishing perturbation at the bottom of helix II. Finally, the loop II and helix III and IV see interactions fairly consistently until a drop in perturbations at the end of helix IV. The most perturbed residues of AcpP included I10 and L15 on helix 1; F28 in loop 1; D35, S36, L37, V43, M44, A45, and E47 in helix 2; A59 of helix 3; and Q66 and A68 of helix 4 are also highly perturbed. Similar to FabG, salt bridges likely form at residues D35 and E47, with E47 likely binding K43′ on FabI. And D35 interacting with R193′. Finally, the residues L37 and M44 on helix 2 form hydrophobic interactions with residues on the FabI interface, demonstrating a binding motif similar to FabG (Fig. 4). Uniquely, the perturbations and docked model of AcpP–FabI show not only the canonical AcpP helix II and III binding to the enzyme but also additional interactions with helix 1. The identified binding region corresponds with previous mutational studies that first identified the AcpP–FabI interface[51]. Titan analysis calculated a $K_d$ of 1.7 ± 1.2 μM with a $k_{off}$ of 8500 ± 2700 $s^{-1}$ and approximately one to one stoichiometry (Table S9 and Fig. S14).

**The TesA *E. coli* thioesterase.** Many organisms utilize a thioesterase to liberate fatty acids from the ACP. In *E. coli*, mature acyl-AcpPs are instead steered directly into other biosynthetic pathways via acyl transfer from AcpP, primarily for phospholipid biosynthesis. However, *E. coli* does possess the thioesterase TesA, which localizes in the bacterial periplasm[59]. Though TesA is not believed to be involved in the terminal step of *E. coli* FAB, it can hydrolyze acyl-AcpP in vitro and has been used as a tool for FAB engineering to increase free fatty acid titer when overexpressed within *E. coli*[60]. Upon titration with TesA, the pattern of C8-AcpP CSPs occurred predominantly in residues different from those perturbed by FAB enzymes (Table S6, Fig. S3, S5 and S10). The perturbations are relatively minor throughout with small perturbations in helix and loop I. There are a larger number of perturbations on helix II, with more than half of the residues being perturbed over the background. Loop II and helix III show

a diminished level of perturbation relative to helix II, this trend continues with few perturbations identified on loop III and helix IV. Overall, the titration by TesA appeared to affect signals from fewer C8-AcpP residues than the other proteins tested. The largest observed residues include loop 1 at S27 and D31; helix 2 at T42, M44, and A45; loop 2 at G52; and loop 3 at T63. Residue D31 appears to interact with R77′ of TesA upon binding. Additionally, D35 appears to interact with the TesA backbone or side chain at S43′. The internal AcpP residues A45 and L42, located within the central hydrophobic core, are both perturbed upon TesA binding. S27 lies in the loop following helix I and near the interface of AcpP and the enzyme, likely experiencing or stabilizing loop motions upon salt bridge formation by D31. M44 appears somewhat distal from the interface near the acyl cargo, although in the case of FabI is part of the interface. T63 appears in the docked model to be positioned to interact with the hydrophobic surface region of TesA (Fig. 4). Titan analysis calculated a $K_d$ of 12.5 ± 7 μM with a $k_{off}$ of 9716 ± 820 $s^{-1}$ (Table S4 and Fig. S15), though these data demonstrate greater error due to the small number and small migration of peaks. The small number of interactions demonstrates that the TesA interface is not optimized for AcpP interactions, further suggesting that it could be engineered to provide a classical interface and increase the interactions and turnover.

**Elucidation of dynamic AcpP•PP interface throughout *E. coli* FAS.** Combining these NMR titrations and docked structures provides a powerful data set of functional PPIs in *E. coli* FAB (Fig. 1A and Fig. S5). When compared against each other, these CSPs demonstrate two important concepts to shape our understanding of AcpP-dependent synthases. Firstly, AcpP•PP interactions are predominantly electrostatic, with the acidic AcpP surface binding to a "positive patch" at the surface of the partner enzyme. However, the majority of the largest CSPs found in these studies correspond to hydrophobic residues (Fig. 3A, B) spanning the interface, acyl pocket, and back of the AcpP. But the data still suggest that electrostatic interface interactions are critical to the protein–protein binding event. Secondly, each enzyme enumerated above binds with AcpP transiently; the weak nature of these interactions is necessary for the "fast" or "fast-intermediate" exchange NMR chemical shifts and agrees with both our presented data and previously known AcpP-binding affinities[32,56]. In both fast and fast-intermediate exchange, interactions between AcpP and PPs are occurring rapidly enough that residues resolve as a single migrating peak on the spectra, rather than two distinct peaks. The titrations effect on lineshape suggests that the interactions are not so rapid that the titrations are happening in "fast exchange". This is also reflected in the TITAN-derived $k_{off}$ rates. These findings demonstrate that recognition between AcpP and its PPs are dynamic processes, driven both by the electrostatic interface and conformational dynamism of the AcpP.

Across the six elongating enzymes tested, half of the residues with perturbations one standard deviation above the mean were at the interface, while the other half of perturbed residues lied in the pocket of AcpP. This is most likely a result of the substrate chain flipping into the PP. Approximately one-third of the largest perturbations, just 10 of 29, are unique to a single partner. More perturbations are shared by three or more of the six enzymes examined than are unique. Each partner, excluding TesA, displays perturbations at the "top" of the acyl pocket, at the start of helix 2 and the helix 3 to the beginning of helix 4. These interactions are likely those responsible for positioning S36 for substrate delivery. TesA is the only enzyme studied, which is known to not be an AcpP FAB partner in vivo but has been demonstrated to have a low level of activity in vitro. Correspondingly, AcpP does not

appear to form the interactions with TesA that are essential for efficient interactions. For other enzymes, it is not unreasonable that AcpP•PP interfaces would predominantly be shared sets of AcpP residues, with a few residues forming unique interactions that contribute to selectivity, given the small size of AcpP and the positively charged binding surfaces of PPs.

## Discussion

Our studies found that re-creating the interface of ACP•PP was achievable across four examples, and the models were in excellent agreement with experimental structures. While our study utilized a system that has a breadth of known information, the active sites of evolutionarily related polyketide synthases are largely conserved, making the inference of a PP's active site possible even without experimentally demonstrated residues. Furthermore, we believe the ability of NMR to appreciate the subtle differences in binding residues on the carrier protein is important to understand selectivity and substrate selection for inhibition or engineering. The combinatorial method synergizes the sensitivity and substrate accuracy of NMR experiments and the ability of informed docking to generate accurate models. This can be used to guide future engineering efforts, leveraging in silico screening for efficiency and economy. Furthermore, given the ability to re-create these interfaces, inhibitor screening should also be possible. The small size, simple electrostatic surfaces, and breadth of background knowledge of partner structures makes carrier protein-dependent pathways ideal systems for computationally guided engineering and inhibition.

Taken together, these CSPs reveal a striking distinction between enzyme classes (Fig. 4A, B). Though AcpP contains only 77 amino acids, the residues involved in each binding are distinct, illuminating how one small protein can interact with dozens of partners. Each PP binding, excluding TesA, induces perturbations at the "top" of the acyl pocket, at the start of helix two and helix three to the beginning of helix four. Although AcpP•PP interactions are typically understood as predominantly electrostatic in nature, half of the largest CSPs correspond to hydrophobic residues (Fig. 4C, D). This represents an evolution in the understanding of type II FAS AcpP•partner recognition, demonstrating that unique residues are used for PPIs with different PPs. We propose a model wherein specific surface interactions are critical for creating allosteric movements within the central channel, triggering the chain flipping event. This may explain the stringent control of reactivity, yet broad range of substrates, necessary for FAB function. The disparate binding motifs found across this iterative pathway provides a compelling model for how a simple <10 kDa protein performs unique interactions for each of six enzymes, while still displaying similarities within classes. This basic model can be extended across the known AcpP interactome, currently at 27 proteins, each of which may demonstrate similarly unique PPIs. Although this constitutes a broad sampling of each enzyme, further study can provide detail into each protein's dynamics and allosteric control. This study provides a foundation with which to expand upon our understanding of PPI driven specificity, PPI redesign, and inhibitor development.

## Methods

**Materials**. The $^{15}$N ammonium chloride used in the labeled growth was purchased from Cambridge Isotopes laboratory. Deuterium oxide ($D_2O$) used in preparation of perdeuterated growth was purchased from Sigma Aldritch. All unlabeled proteins were grown on Luria broth from Teknova.

**General PP purification protocol**. All PPs were generated through overexpression in *E. coli* BL21 (DE3). Cells and grown in LB media. Cells were grown in the presence of 50 mg/L kanamycin before induction with 1 mM IPTG at $OD_{600} = 0.8$ and incubated at 18 °C for 12–18 h. Cells were pelleted in a JLA-8.1 rotor at 800 RCF. Cells were re-suspended and lysed by sonication in 50 mM Tris-HCl

(pH 7.4), 250 mM NaCl, and 10% glycerol. The lysate was then spun at 10,000 RCF in a JA-20 rotor for 1 h to pellet the membrane and insoluble materials. Proteins were purified using Ni-IMAC (Bio-Rad) after a 30-min batch-binding time rotating at 4 °C. The general protocol used two solutions, a wash of 40 mL of lysis buffer followed by a wash of 40 mL lysis buffer with 15 mM imidazole added. This was followed by three 5 mL elutions with lysis buffer containing 250 mM imidazole. Unless stated in the purification specifics below this was the method used in all purifications. After purification proteins were concentrated to ~2 mL and purified using size exclusion chromatography on a Superdex 75 column. The PP was purified into NMR buffer and concentrated to the concentrations listed below before addition to the tube for NMR.

AcpP was grown on a His-tagged pET-22b vector in *E. coli* BL21 (DE3) cells. Cells were grown at 37 °C in M9 minimal media containing 1 g of $^{15}$N NH$_4$Cl and 8 g of glucose. Perdeuteration was achieved by growing *E. coli* in increasing ratios of $D_2O$. Starting by growing 5 mL overnight in 25% $D_2O$/75% $H_2O$, this was used to inoculate 5 mL cultures which were 50% $D_2O$/50% $H_2O$. This same technique was used to inoculate 75% $D_2O$, 90% $D_2O$, and finally 100% $D_2O$ starters. This starter was used to inoculate the liter of deuterated media. Once the growth reached an $OD_{600}$ of 0.8 they were induced with 1 mM IPTG and allowed to grow for an additional 4 h at 37 °C.

Following purification AcpP was dialyzed overnight into 50 mM Tris, 250 mM NaCl, and 1 mM DTT buffer, in order to remove the imidazole before subsequent reactions. AcpP was first prepared as uniformly apo by reaction with *Pseudomonas aeruginosa* ACPH in a solution with 5 mM MgCl$_2$, and 0.5 mM MnCl$_2$, and 1 mM DTT. This reaction was performed overnight at 37 °C. Apofication was confirmed by conformationally sensitive UREA-PAGE. Following this, loading was performed using three *E. coli* biosynthetic enzymes CoaA, CoaD, and CoaE and the *Bacillus subtilis* SFP. The reaction is performed with 12.5 mM MgCl$_2$, 10 mM ATP, 0.1 μM CoaA, 0.1 μM CoaD, 0.1 μM CoaE, 0.2 μM Sfp, 0.02% Triton X, 0.01% Azide, and 0.1% TCEP. The reaction was performed overnight at 37 °C, with loading confirmed by conformationally sensitive UREA-PAGE. Stable C8 acyl loaded ACP analogs were achieved through loading of an octanoyl pantethenamide probe.

**Purification and sample preparation of FabF**. ACP was concentrated to 3.87 mg/mL using Amicon Ultra-15 3 kDa centrifugal filters, and a Nanodrop was used to measure concentrations with the extinction coefficient 1490 M$^{-1}$ cm$^{-1}$. FabF was concentrated to 10.1 mg/mL using Amicon Ultra-15 10 kDa spin filters, using the extinction coefficient 25,900 M$^{-1}$ cm$^{-1}$. These were used to create a 0.042 μM AcpP zero-point sample and a 0.042 μM AcpP 0.0837 μM FabF saturated sample. FabF was purified for titration the day before the experiment, ensuring a 'fresh' sample for maximum stability. Perdeuterated AcpP was used to boost signal from sensitivity lost due to the titrated PP. This also lowers the concentration of PP necessary to accommodate stability concerns. The proteins were purified into a 10 mM potassium phosphate pH 7.4 buffer with 0.5 mM TCEP and 0.1% NaN$_3$ and any buffer added to the tubes was taken from the same FPLC buffer solution. The total volume of both the zero point and saturated samples was 450 μL, and 50 μL of $D_2O$ was added to both tubes for locking.

**Purification and sample preparation of FabI**. ACP was concentrated to 2.38 mg/mL using Amicon Ultra-15 3 kDa centrifugal filters, a Nanodrop was used to measure concentrations with the extinction coefficient 1490 M$^{-1}$ cm$^{-1}$. FabI was concentrated to 22.1 mg/mL using Amicon Ultra-15 10 kDa spin filters, using the extinction coefficient 15930 M$^{-1}$ cm$^{-1}$. These were used to create a 0.0669 μM AcpP zero-point sample and a 0.0669 μM AcpP 0.2689 μM FabI saturated sample. This high equivalent concentration was used to ensure that there would be at least a 1:1 ratio of ACP: FabI tetramer. The FabI was purified the day before the experiment and concentrated to the high molarity necessary the morning of the experiment in order to ensure the sample was as stable as possible. Perdeuterated AcpP was used to boost signal form quenching. The proteins were purified into a 10 mM potassium phosphate pH 7.4 buffer with 0.5 mM TCEP, 0.5 mM NAD+ and 0.1% NaN$_3$ and any buffer added to the tubes was taken from the same FPLC buffer solution. The total volume of both the zero point and saturated samples was 450 μL, 50 μL of $D_2O$ was added to both tubes for locking.

**Purification and sample preparation of FabG**. ACP was concentrated to 2.25 mg/mL using Amicon Ultra-15 3 kDa centrifugal filters, a Nanodrop was used to measure concentrations with the extinction coefficient 1490 M$^{-1}$ cm$^{-1}$. FabG was concentrated to 11.21 mg/mL using Amicon Ultra-15 10 kDa spin filters, using the extinction coefficient 11460 M$^{-1}$ cm$^{-1}$. These were used to create a 0.0538 μM AcpP zero-point sample and a 0.0538 μM AcpP 0.2199 μM FabG saturated sample. This high equivalent concentration was used to ensure that there would be at least a 1:1 ratio of ACP:FabG tetramer. The FabG protein was purified the day before titration and concentrated the morning of the experiment to ensure a stable sample for the experiment. Perdeuterated AcpP was used to boost signal form quenching. The proteins were purified into a 10 mM potassium phosphate pH 7.4 buffer with 0.5 mM TCEP, 0.5 mM NAD+ and 0.1% NaN$_3$ and any buffer added to the tubes was taken from the same FPLC buffer solution. The total volume of both the zero point and

saturated samples was 450 μL, and 50 μL of $D_2O$ was added to both tubes for locking.

**Purification and sample preparation of TesA**. ACP was concentrated to 6.5 mg/mL using Amicon Ultra-15 3 kDa centrifugal filters; a Nanodrop was used to measure concentrations with the extinction coefficient 1490 $M^{-1}cm^{-1}$. TesA was concentrated to 3.2 mg/mL using Amicon Ultra-15 10 kDa spin filters, using the extinction coefficient 40450 $M^{-1}cm^{-1}$. These were used to create a 0.0538 μM AcpP zero-point sample and a 0.0699 μM AcpP 0.104 μM TesA saturated sample. TesA was prepared the day before the experiment in order to have a fresh and stable sample for the experiment. Due to the small size of TesA, perduteration was likely not necessary but was maintained for consistency. The proteins were purified into a 10 mM potassium phosphate pH 7.4 buffer with 0.5 mM TCEP, and 0.1% $NaN_3$ and any buffer added to the tubes was taken from the same FPLC buffer solution. The total volume of both the zero point and saturated samples was 450 μL, and 50 μL of $D_2O$ was added to both tubes for locking.

**NMR methods**. Experiments were performed on a Bruker 800 MHz spectrometer equipped with a cryo-probe at the UCSD Biomolecular NMR facility. Previous assignments[35] of the C8-AcpP were used to assign the backbone peaks on the HSQC for all experiments. Each experiment was performed at 37 °C, with each titration including at least five titration points in order to observe the full movement of peaks. CSPs were quantified using the formula[32]

$$CSP = \sqrt{\frac{1}{2}\left[\delta_H^2 + (\alpha \cdot \delta_N^2)\right]}$$

An α value of 0.2 was used in all CSP calculations. Titrations were performed by the preparation of an initial saturated and zero-point sample. To create intermediary samples for the titration these two samples were mixed. AcpP is an extremely stable protein, with no denaturation observed. However, in some cases a degree of PP crash was seen in the later titrations. To mitigate this when not in the spectrometer samples were kept refrigerated at 4 °C and the spectra were collected one after another over the course of ~12 h. Through this cautious approach and fresh PP preparation we were able to collect all saturated samples with no observed crashed partner, though in the case of FabF and FabG a small amount of crashed protein was observed for the final "middle" spectra's sample (3.0 equivalents in FabG and 1.0 equivalents in FabF). All HSQC were acquired with a 1.5 s recycle delay and 2048 data points in the spectra. Spectra were processed using in NMR Pipe[61] and NMRFAM-SPARKY[62]. Spectra were visualized and all NMR spectra figures were generated in SPARKY. CSP calculations and the CSP heat map were generated in the Matplotlib python utility[63]. NMR titration data were further analyzed using the TITAN 2D lineshape analysis program. For the analysis in all cases a flexible stoichiometry model was used. Both to allow for examining the stoichiometry of the interactions and in order to allow the concentration of the AcpP to vary in the calculation. This was done because AcpP has a very low extinction coefficient, making quantifying exact AcpP concentrations difficult. In the analysis five titrations steps were used in each analysis. The calculations were performed by selecting each region of interest on the spectra. Following this the initial fitting was performed, with the parameters first estimated at 10 μM with a $k_{off}$ rate of 5000 $s^{-1}$. After the fitting each peak chosen was hand checked, in order to ensure that the peak had been properly fit. Though the TITAN program self fits peaks, in more crowded regions of the spectra the program can improperly fit the wrong peak. After the initial fitting jackknife error analysis was performed, this was done in order to verify the program ran without issue. Improperly fit peaks were identified in the more rapid bootstrap error analysis, peaks which had very small migrations or which migrated into other peaks displayed high error and were hand chosen to not be fitted. In this way Jackknife analysis was used to identify user error and occasional problematic peaks before the much longer bootstrap analysis was run. After these problematic peaks were hand checked a final bootstrap analysis was performed, in all cases 300 steps were performed in the analysis. The results of this bootstrap analysis were used as the error in reporting values, a set of simulated and "real" peaks are presented for each analysis below. A selection of two contour plots and two 3D contour plots for each analysis is supplied. It was observed that the FabG, FabF, and TesA resulted in greater error in the analysis. We suspect the error in FabG, FabF, and TesA is due to the instability of the PP seen in the final titration step. FabG in particular had all of the components at extremely high concentrations in order to ensure saturation.

**Docking method**. In the case of FabF, FabB, FabI, FabG, FabA, and TesA structures for the PP were acquired from the protein data bank; 2GFW, 1G5X, 4CV3, 1Q7B, 1MKB, and 1IVN were used respectively. For FabZ the only structure available is 6N3P, a crosslinked crystal structure. As such this was used but AcpP was deleted. 2FAD was used as a starting crystal structure, with an additional carbon added in ICM to elongate the 7 carbon acyl chain and create a C8 acyl chain. All PPs were used in the subunit structure which it adopts in solution, especially given that in many cases AcpP binds multiple subunits of a dimer or tetramer. Specifically: FabI and FabG were docked as tetramers. FabA, FabZ, FabF, and FabB were docked as dimers. TesA was docked as a monomer.

**Preparation of PDB proteins for docking simulations**. Before docking the proteins were prepared by solvation and minimization. The AcpP and all PPs were solvated with the ICM quickflood procedure in order to generate a water box. After this the proteins were minimized in ICM to optimize side chain orientations and hydrogen bonding with the water box and any ligands. This was performed by running the optimizeHbonds and optimize HisProAsnGlnCys protocols in order to form more solution relevant conformations of the residues. After this the cofactors, including the AcpP substrate, were deleted and the proteins were docked. However, the structural effects of the AcpP substrate remain reflected in the docking, with the acyl pocket remaining during the simulation.

**ICM docking—"Informed" and "uninformed" procedures**. Docking was performed both with and without focus residues using the ICM fast Fourier transform protein docking protocol. The "informed" docking procedure was performed by specifying experimentally known interacting residues on the AcpP and PP. Explicitly, the "uninformed" docking jobs are performed by docking the exact same input structures, without focus residues. In more detail, given the history of mutational study in *E. coli* FAB we leveraged studies which mutated the AcpP interface and saw diminished activity. As an example in the docking of FabG the work of Price et. al., 2004 was used. In their paper they identified the region of NADP binding and proposed an interface. R15 was identified at the edge of the NADP pocket. The R129 and R172 were not chosen because they perform a more complicated docking with the AcpP from the adjacent chain. However, the final model generated did dock such that these two identified residues were interacting with AcpP. This same methodology is very broadly applicable in carrier protein-mediated biosynthesis, given the depth of the literature. For the calculations interface residues on the AcpP were chosen by selecting the largest perturbations whose position was such that they were likely hydrogen bonding at the interface. Docking poses were sorted by lowest energy, it was examined if scoring based on Van der Waals or electrostatic interactions specifically would yield more accurate structures. But we noted that the ICM energy scoring function performed best. Table S7 displays the RMSD of the top 10 poses generated by the informed and uninformed docking jobs.

**Docking with ClusPro, HADDOCK, and Rosie**. It should be noted that the methodology used is not unique to ICM, as such we have performed three docking experiments with commonly available online docking servers. Docking was performed with three common servers: ClusPro[42–44], HADDOCK[45,46], and Rosie[47,48] with the PP FabF. The results are reported in Table S8, comparing the top three structures generated from the servers. Models created by ICM, ClusPro, HADDOCK, and Rosie were compared to the crosslinked crystal structures through two methods. The first was a full alignment in Pymol, this yielded good values but often global changes in the PP and AcpP either upon crosslinking or due to differences in substrate or crystallization conditions appeared to have altered the backbone distant from the interface. To specifically look at the interfaces between the two enzymes the atoms within 5 Å of the interface were selected for both the docked and crystal structures. The two interfaces were then superimposed upon one another using the Pymol super command and the value was reported without deletion. Pymol was used in all structural visualizations. For the comparison study using online servers the input files used in all docking, post solvation, and minimization were collected. For HADDOCK docking both the ICM prepared files and the raw PDB files (2FAD and 2GFW) the inputs were loaded into the server and active residues were defined as 65 and 616 on FabF and 35 and 39 on AcpP, the same focus residues in the ICM study. Passive residues were defined within 6.5 Å of the active residues as was suggested by the program. As the study is meant to sample the most accessible components of the method the "EASY" access level account was used. Meaning all parameter settings were set to default. The results were ranked using the standard energy scoring metric in HADDOCK and benchmarked by alignment to the crosslinked structure. The Cluspro docking was the simplest method performed for this study; the same input structures were used as the HADDOCK. After loading in the structures the attraction residues were set to be the same as in the HADDOCK and ICM jobs. Both the electrostatic and balanced scoring functions are presented. The Rosie server was run with the Docking2 refinement utility, as this utility requires an input structure the Cluspro best ranked "Electrostatic scoring" docked file was used. We felt this represented a second refinement step which could easily be taken by other groups after the rapid Cluspro docking. However, we would advise caution when performing refinements of ClusPro docking. Great care should be taken to ensure that the starting structures are an accurate starting structure and that Rosie is not optimizing an incorrect interface. We feel this demonstrates the ability of many utilities to re-create AcpP partner interfaces. As well it demonstrates the importance of caution when evaluating docked structures from Cluspro. HADDOCK performed extremely well when supplied with properly prepared starting structures, but we feel this stands as additional evidence that preparation of the proteins to re-create a solvated structure is important. As like ICM HADDOCK performed poorly when provided with structures straight from the PDB.

**Statistics and reproducibility**. NMR analysis was performed on single sets of $^1H$-$^{15}N$ HSQC experiments. Chemical shift perturbations were calculated using the

supplied equation and $\alpha$ value. TITAN analysis was performed according to the method outlined by the developers. Calculation of error was performed using only the recommended bootstrapping method, with 300 steps of calculation. RMSDs were calculated using the Pymol align command, in all cases of alignment the 10 lowest energy states are reported. Starting structures were taken from publicly available repositories.

**Reporting summary**. Further information on research design is available in the Nature Research Reporting Summary linked to this article.

## Data availability

The docked models that support the findings are available on the ModelArchive. TesA: ma-kqhmx, FabI: ma-k4kpl, FabG: ma-zujwj, FabA: ma-gu2rg, FabB: ma-9dj6n, FabF: ma-gpsyu. The NMR titration data are available on the BMRB under codes: 50554, 50559, 50560, and 50561. Detailed NMR peak assignments are shown in Supplementary Data 1–4. Top ranked PDB structures from docking simulations are provided in Supplementary Data 5. All other data are available from the authors by request.

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

## Acknowledgements
We thank S. Opella, X. Huang, and the UCSD biomolecular NMR facility for support and guidance. This work was funded by NSF IOS-1516156 and NIH R01GM095970.

## Author contributions
T.B. performed NMR experiments and developed docking protocol. R.A. provided support and guidance, and A.P. provided assistance for docking. T.S. aided preparation of matplotlib and pymol scripts. D.J.L. provided NMR data and interpretation. M.A.Y. assisted protein preparation and analysis. M.D.B. supervised the study and analysis. All authors participated in manuscript preparation and editing.

## Competing interests
The authors declare no competing interests
