## [Peer Review File · Communications Biology]

Reviewers' comments:

Reviewer #1 (Remarks to the Author):

The major strength of this manuscript is the elucidation of different patterns of chemical shift perturbations (CSPs) of backbone ¹H-¹⁵N NMR signals from [U-¹⁵N]-C8-Acp when titrated with six different binding partners (Fig. 4B). Some possible interacting residues of the partner proteins were deduced from structural models of complexes created by in silico docking. The NMR titration data were analyzed to obtain crude measures of dissociation constants and off rates for the complexes. These will be of interest to specialists.

In its current state, the manuscript exhibits several weaknesses.

1. Sufficient data are not provided for reproducibility, ideally as deposited in publically accessible databases.

The full NMR titration data for all complexes should be deposited in the BMRB. The starting chemical shifts and modeled chemical shifts on saturation (provided in the supplemental tables) are insufficient. The 3D coordinates of the hybrid structural models of the complexes should be deposited in PDB-DEV, the archive of the Protein Data Bank recently set up for such data.

2. Some procedures are not described clearly or completely as shown by examples.

(line 107) "To maintain the applicability of the method to systems without crosslinked structures all docking was performed with an apo partner protein structure and the 2FAD crystal structure." This would be clearer as: "Docking simulations were carried out between the x-ray structural model of heptanoyl-AcpP-C7 (PDB 2FAD, missing ref.) to which a methylene was added to the acyl chain simulate C8-AcpP and crystal structural models of binding partner proteins from which cofactors had been removed."

(Line 125) "Data showed that all the enzymes tested had docking orientations which allowed for chain flipping of the substrate, demonstrating the ability of informed calculations to filter out nonproductive complexes." Did chain flipping occur in any of the models? What is meant by "allowed for chain flipping"?

It is axiomatic that RMSD values of simulations decrease when more restraints are imposed. How is the lack of improvement in the RMSDs of the model of the FabA complex with added restraints to be explained? (line 131)

(line 267) "Our experiments found that recreating the interface of ACP•PP was feasible and in agreement with experimental structures of crosslinked complexes." Wasn't only one crosslinked structure examined? If so, then "structure of a crosslinked complex."

(line 466) Figure 3B are the stick representations for only one model? If so, which model?

Supplement under NMR:

"The analysis was seen to on occasion identify very distant regions of the spectra," What does this mean?

"After the initial fitting jackknife error analysis was performed, individual peaks which displayed very high error were hand checked." Please describe what is meant by "high error" and how it is manifested.

Supplement page ? Rosie appears to yield the best structural model for the complex as compared to the crosslinked X-ray model. However, this approach was not used with the other complexes, but instead a poorer performing method. Why?

Table S1: It is unclear to the reviewer how these data were obtained. A description is needed either in the text or in a caption to the table.

Table S7: The data lack specified units.

Table S9: How is the outlier n value for the FabG complex to be explained? Would it have not been possible to quantify the relative concentration of the components of the complex by amino acid analysis? This would have improved the analysis.

3. Figures are incompletely or incorrectly referenced in the text and supplement.

In the text, Figures S9, S11, S12, S13, S14, and S15 are not referenced, nor are Tables S8 and S9. In the supplement, none of the supplemental figures are referenced in the Supplementary Methods, and of the nine supplemental tables, only Table S7 is referenced.

(line 187) "The most perturbed most residues (Table S5, Fig. S7) on AcpP were: N25 and F28 of loop 1; D35, S36, L37, T39, V40, E47, and F50 on helix 2; and T64, Q66, and A68 on helix 3." Probably better as: "The residues of C8-AcpP exhibiting the largest CSPs (Table S5, Fig. S2) were: N25 and F28 on loop 1; D35, S36, L37, T39, V40, E47, and F50 on helix 2; and T64, Q66, and A68 on helix 3." [note that the figure was incorrectly cited]

(line 194) "and a one to one stoichiometry (Table S4)" correct to: "and a one to one stoichiometry (Table S9)" [incorrect table cited]

(line 211) "(Table S4)" should be "(Table S9)"

(line 218) "Perturbations predominantly occurred in unique locations compared to FAB enzymes (Table S6, Fig. S10)" would be clearer as "Upon titration with TesA, the pattern of c8-AcpP CSPs occurred predominantly in residues different from those perturbed by FAB (Table S6, Fig S3). [note incorrect figure citation]

4. The text is poorly written in places as shown by examples.

Pages in the supplement are not numbered.

Examples of non-sentence (line 93): "With distinct residues involved in specific classes of enzyme binding, and internal residues that are likely important to the mechanistic process of chain flipping."

And (line 248) "Where AcpP and partner proteins are binding and dissociating fast enough to resolve as a single migrating peak on the spectra, rather than two distinct peaks."

(line 146) "FabF displays perturbations throughout the AcpP, beginning at the end of helix I" would be more accurate as, "Upon titration with FabF, residues from C8-AcpP display chemical shift perturbations beginning at the end of helix I"

(line 149) "the visible residues are perturbed nearly throughout the AcpP" better as "signals assigned to residues nearly throughout C8-AcpP were perturbed"

(line 154) "(residues identified from the models will be denoted by apostrophes throughout)" would be clearer as "(residue designators for the partner proteins in the complexes will be denoted by primes)"

(line 161) "FabB performs the same ketosynthase reaction" as what?

(line 163) "Overall FabB displays a slightly less broad set of CSPs than FabF" Again, signals from C8-AcpP are displaying the CSPs.

(line 169) "PPIs of AcpP with FabB contain a large number of predominantly acidic residues at the interface: D35, D38, and E60" Awkward sentence.

(line 172) "There is a small number of internal hydrophobic CSPs from L15 and L42" probably would be better as "A small number of hydrophobic residues, such as L15 and L42, exhibit CSPs"

(line 197) "Perturbations of AcpP by interaction with FabI are distributed throughout the ACP." Might be clearer as: "Upon interaction with FabI, NMR signals from residues throughout C8-AcpP exhibited CSPs."

(line 223) "Overall TesA appears to effect less residues than the other proteins tested." Probably better as: "Overall, the titration by TesA appeared to affect signals from fewer C8-AcpP residues than the other proteins tested."

(line 418) "Figures S1 to S11." Should be "Figures S1 to S15."

(line 419) "Tables S1-S6." Should be "Tables S1-S9"

(line 451) Equation: suggest removing the "x" which looks like a variable as in Supplement page 2. Supplement page ? "who's" should be "whose"

Supplement page ? "Van Der waal's" should be "Van der Waals"

Supplement: caption to Figure S4: Remove "(Larger versions in Figure S4)."

5. Literature is not cited in several places where it would be appropriate as shown by examples.

(line 143) C8-AcpP was titrated with increasing concentrations of the FabF ketosynthase (Fig. 2a) and compared to recently published data of C8-AcpP titrated with FabB (missing ref.).

Supplement page ?: "ICM", "Cluspro", "HADDOCK", etc. should be referenced.
Figure S6 needs a reference. Are the data in BMRB?

Reviewer #2 (Remarks to the Author):

This manuscript describes an exhaustive study on protein-protein interactions in fatty acid biosynthesis, by integrating NMR and docking simulations. The combination of experimental and computational analysis provides interesting mechanistic insights into the protein interactions in this pathway. This is a good contribution to the field, and I have only some minor comments.

Minor comments:

Line 98: "...the residues on the PP that mediate binding cannot be determined by this process...". Can you elaborate more on this? Does it refer to structural/energetic/dynamics contribution of residues?

- Most of the methodological details are described in the supplementary material (e.g. docking protocol, how proteins were minimized in water, which are the "expanded calculations"). However, it took me a while to figure out that. For a broad reader, it would be helpful to include explicit referral to the supplementary material in the main text.

- They compared the docking model with available cross-linked structures. Which is exactly the purpose of this? Is it because the cross-linked structures are not representing the biological complex? Or is the purpose to check the docking performance? If it is already mentioned, I missed it, so it should be more clearly stated.

Reviewer #3 (Remarks to the Author):

1. Brief summary of the manuscript

The manuscript presents studies of the fatty acid synthase type II acyl carrier protein AcpP interacting with six of its partners, as seen through a combination of NMR experimental data and docking algorithms. The objective is to provide a protocol for determining similar docking models in related systems, i.e. for carrier proteins that interact with multiple partners. The method is illustrated with six partners of AcpP and compared with existing results from complementary techniques.

2. Overall impression of the work

The significance of the system is high, and so is the significance of the authors' objectives as predicting the impact of mutagenesis on such protein-protein interactions could offer several promising applications. The proposed method could be a valuable tool to accelerate progress in the field. Before I can make a complete assessment and validate both the authors' interpretation and the usefulness of their protocol, I would need some (I believe straightforward) revisions. The manuscript has potential but the authors need to articulate better the novelty of their protocol and its associated findings. Currently, the manuscript hesitates between being a method paper (as advertised, but then the format does not match) and a paper showcasing how well-established NMR experiments provide information that is complementary to other techniques.

A large number of omissions or confusion indicates that submission may have been premature.

3. Specific comments, with recommendations for addressing each comment

I was looking forward to reading this manuscript and learn about a protocol that could provide docking models for proteins involving unusual chemical modifications (AcpP and its cargo interacting with its partners) and to see how NMR could fill gaps in existing mechanisms of type II FAS protein interactions. Although the authors should be commended for their effort, a few issues need to be resolved.

There may be a catch 22 to resolve here. The reason that NMR brings new information about mechanisms, as the authors describe well, may limit applications to docking. Traditionally, NMR-assisted docking relies on the assumption that CSPs of surface residues report on molecular binding through contacts at the binding sites. Here, for AcpP, many molecular interactions are at play and NMR is sensitive to all of them. Notably, the PP and its tethered load must disengage the CP core before interacting with partner proteins. It is unclear to me how the signals of residues at the surface of AcpP may be impacted by the trajectories of the phosphopantetheine arm and its cargo as proteins engage. How are these contributions to CSPs delineated from those stemming from PPI when using constraints for docking? Is it possible that some differences in CSP fingerprints for different CP partners reflect changes in phosphopantetheine behavior rather than alternative interactions? If so, how did the authors address the contributions of phosphopantetheine when creating docking models? From the abstract and introduction, I anticipated that the manuscript would precisely address these issues as a new docking protocol is promised.

Please include a Method section and expand on what you currently provide in the SI. What is the buffer for modification of AcpP and the temperature and timing of reaction, what are the concentrations of reagents? etc., etc. How were NMR titrations conducted? Did you mix saturated and reference samples to create the points in between? Was the data acquired chronologically (1:0, 1:X, ..., 1:max) or scrambled in time? How was the integrity of NMR samples monitored (e.g. overlay of 1Ds at the beginning and end of titrations/acquisition or SDS-PAGE or MS before and after NMR)?

As you present a new method, please provide a detailed docking protocol that simultaneously highlights the novelty of your docking procedure and that can be used by readers in a reproducible manner. The section presented in the SI most likely provides sufficient details, but it is currently provided with little effort for clarity and flow. If your paper is in part about the method, this section should not be in the SI.

It seems the authors hesitate between using cross-linked structures as references for validating docking models and explaining that docked models differ because the methods are different. Please resolve.

References to SI material are in general wrong. SEC traces and SDS-PAGE are mentioned when describing interactions, and tables of shifts are referred to when describing Titan analysis.

Tables have no captions.

You report a K_d of 20 +/- 40 μM .

Although the quality of figures in SI is too poor for rigorous assessment, your data appears to often

display intermediate/fast exchange and not fast exchange.

"Perdeuterated AcpP was used to boost signal from quenching." Reword (not the right spectroscopy).

Please relate explicitly the orientations of AcpP that are presented, in particular for surface representations (preferably in captions). I am guessing 2E has the same orientation as 2D with a reduced scale and that 4B uses the same orientations.

About the SI. The figures in the supporting information need to be revised for clarity; I managed to assess the take-home points but the data cannot be used and interpreted in full by a reader:

- In all spectra, increase the line thickness for the contours. Then verify that the choice of colors makes them legible. Currently, the lines are so thin that we cannot distinguish colors. Also, change the font for assignments to sans-serif and make them bold or increase the size: I had to guess quite a few.

- Verify that all axes labels are legible. This is not the case for CSPs nor SEC traces.

- The purity of all samples in general is well demonstrated. For FabI, Figure S4, there is a shoulder in the SEC, and the SDS-PAGE is set for sensitivity rather than resolution: if you have a diluted SDS-PAGE to show that fractions 10-13 do not contain two overlapped bands, show it as well. Otherwise, please discuss the shoulder in SEC. TesA is probably fine given its SEC but the SDS-PAGE is psychedelic and of not much use. Please replace.

- Please verify that the captions in SI are clear. The following sentences are unclear: Figure S11 : "Due to the experiments presented octanoyl-loaded AcpP was purified fresh for each titration.", Figure S2 "The Y-axis is held at the 0.18 for all titrations performed." As mentioned, all items should have captions. I am not sure how to interpret S7 for example.

About clarity (throughout manuscript): Some wording is misleading or confusing.

"This work reveals the molecular basis of six discrete binding events responsible for [...]" and similar statements suggest that there was no information about AcpP in complex with partners. There are structures for all systems (maybe cross-linked, maybe with holo forms, but they exist), and two NMR titrations have already been published by the authors.

Similarly, line 103 "that have eluded experimental structural characterization" must refer to a long term objective for other interactions than those presented here. The statement added to my confusion.

I am not sure what is combinatorial in the approach that the authors present.

Line 268 "the ubiquity of the active sites of modular synthases is such that even when partner residues are not known the correct binding site can be inferred." You need to develop your thoughts. This sentence could mean that active sites are conserved to the point that there is no need to identify them.

Another round of proofreading is needed:

Line 51: "[...] enzyme players (Fig. 1a). While presenting a combinatorial [...]"

line 90-91: "[...]interactions, both in carrier protein-howemediated biosynthesis [...]"

I recommend working on the choice of references to more accurately assign central findings to the original papers. Currently, it appears that ACP substrate sequestration was a rather recent finding when it isn't. The choice of refs 34 and 35 to depict general applications of CSPs for PPI studies is inappropriate. Either cite again the reviews you mention (which are indeed a great choice) or cite

seminal work mentioned in these reviews. Also, Markley and Co., as well as Prestegard and Co. or Kim and Co. or Yang and Co. made important contributions to ACPs mechanisms and demonstrated well the use of NMR for such studies. Citing these efforts may simultaneously restore the historical accuracy of ACP studies and provide examples of NMR applications.

"findings of this study are available from the corresponding author upon reasonable request" either define "reasonable" or remove.

Response to Reviewer Comments COMMSBIO-20-2681-T

Reviewer #1

Item 1.1.A. Sufficient data are not provided for reproducibility, ideally as deposited in publically accessible databases.... The full NMR titration data for all complexes should be deposited in the BMRB. The starting chemical shifts and modeled chemical shifts on saturation (provided in the supplemental tables) are insufficient.

Thank you for this suggestion. We have now submitted our NMR data to the BMRB as suggested. We have included the DOI for our entry into the data availability statement.

Item 1.1.B. The 3D coordinates of the hybrid structural models of the complexes should be deposited in PDB-DEV, the archive of the Protein Data Bank recently set up for such data.

Thank you for this suggestion. For full transparency, we have submitted our data to the ModelArchive. We have included all of the DOI information for the models in the Data availability statement.

Item 1.2.A. Some procedures are not described clearly or completely as shown by examples. (line 107) “To maintain the applicability of the method to systems without crosslinked structures all docking was performed with an apo partner protein structure and the 2FAD crystal structure.” This would be clearer as: “Docking simulations were carried out between the x-ray structural model of heptanoyl-AcpP-C7 (PDB 2FAD, missing ref.) to which a methylene was added to the acyl chain simulate C8-AcpP and crystal structural models of binding partner proteins from which cofactors had been removed.”

This is definitely a more straight-forward way to explain the procedure. We have changed the text to, “Docking simulations were carried out between the x-ray structural model of heptanoyl-AcpP-C7 (PDB 2FAD, missing ref.) to which a methylene was added to the acyl chain simulate C8-AcpP and crystal structural models of binding partner proteins from which cofactors had been removed.”

Item 1.2.B. (Line 125) “Data showed that all the enzymes tested had docking orientations which allowed for chain flipping of the substrate, demonstrating the ability of informed calculations to filter out nonproductive complexes.” Did chain flipping occur in any of the models? What is meant by “allowed for chain flipping”?

We were referring to the analysis that our models all were positioned, such that the most stable complex we found were nearby the known active site. We have changed the selected text: “All the enzymes tested had docked orientations which allowed for chain flipping of the substrate, based on the position of serine 36 and known active sites of the partners. Demonstrating the ability of informed calculations to filter out nonproductive complexes.” We hope that this helps clear up what was meant and appreciate the opportunity to make intent more easily understood.

Item 1.2.C. It is axiomatic that RMSD values of simulations decrease when more restraints are imposed. How is the lack of improvement in the RMSDs of the model of the FabA complex with added restraints to be explained? (line 131)

We agree. It seems that FabA is well modeled without constraints, with the uninformed models being the only ones which align with an average under 10Å. We believe the comparably poorer docking is due to the greater structural difference between the FabA un-crosslinked and crosslinked structures when compared to the other partner proteins. In the full alignments, the partner protein alone was more different than in previous cases. We also believe this is why the FabZ was our most accurate model, given that we had to use the crosslinked crystal structure with the AcpP deleted. It is difficult to say with total confidence why the FabA would perform better than the other partners in “uninformed” calculations. However, appreciate this question and have added the following to the text in order to explain some of our thought on the simulations, “Furthermore, it should be noted that the crosslinked and apo partner proteins have differing structural similarity. With the FabF structure 1.3Å different between crosslinked and uncrosslinked, FabB 3.5Å different between crosslinked and uncrosslinked, and FabA 4.6Å different between crosslinked and uncrosslinked structures.”

Item 1.2.D. (line 267) “Our experiments found that recreating the interface of ACP•PP was feasible and in agreement with experimental structures of crosslinked complexes.” Wasn’t only one crosslinked structure examined? If so, then “structure of a crosslinked complex.”

That is correct. This was an error in sentence structure. We meant to state that we have demonstrated the ability to accurately recreate their interface for multiple partner proteins. We changed the text to, “Our studies found that across 4 examples of recreating the interface of ACP•PP was feasible and the models were in agreement with experimental structures.” We hope this will be a more eloquent way to describe the work and thank you for your help.

Item 1.2.E. (line 466) Figure 3B are the stick representations for only one model? If so, which model?

Supplement under NMR:

“The analysis was seen to on occasion identify very distant regions of the spectra,”
What does this mean?

In more crowded regions of the spectra we saw that TITAN would, on occasion, misidentify a nearby peak as the final saturated point. In these instances, we hand-checked the assignment after the fitting in order to ensure that the peaks which were fit were correct. Thank you for pointing out the poor explanation, as we had some difficulty clearly describing this procedure. The new methods section should now clarify this point.

Item 1.2.F. “After the initial fitting jackknife error analysis was performed, individual peaks which displayed very high error were hand checked.” Please describe what is meant by “high error” and how it is manifested.

This is related to our difficulty with describing the method. We found that on peaks which the program had difficulty fitting, such as those where two peaks migrated into one another, could be identified in the error analysis. Even if the saturated point could be correctly identified, as seen when checked by hand, peaks which migrated into one another resulted in very broad signals which could not be well fit in the analysis. We have revised this section of the methods to more properly explain this protocol.

Reveiwer 1.2.G. Supplement page ? Rosie appears to yield the best structural model for the complex as compared to the crosslinked X-ray model. However, this approach was not used with the other complexes, but instead a poorer performing method. Why?

We had reservations about utilizing the Rosie method for complexes without crosslinked structures to compare. The Rosie server refinement works from an existing structure,

and the ClusPro global docking performed with differing results. This raises the question of how to ensure the initial ClusPro pose chosen is accurate. Furthermore, the structural preparation, which we identified as being very important, was performed in ICM, though we expect that similar preparations would be possible through other programs. To address this issue, we have added the following text, “However, we would advise caution when performing refinements of ClusPro docking. Great care should be taken to ensure that the starting structures are an accurate starting structure and that Rosie is not optimizing an incorrect interface.”

Item 1.2.H. Table S1: It is unclear to the reviewer how these data were obtained. A description is needed either in the text or in a caption to the table.

We have added the following caption, “Alignments are generated from ICM docked models and aligned to crosslinked structures. The “informed” ACP residues are supplied from NMR titration data and the “informed” partner residues were taken from published data on essential electrostatic residues when possible.”

Item 1.2.I. Table S7: The data lack specified units.

We have specified that the alignments are in angstroms in the text.

Item 1.2.J. Table S9: How is the outlier n value for the FabG complex to be explained? Would it have not been possible to quantify the relative concentration of the components of the complex by amino acid analysis? This would have improved the analysis.

We have invested a significant effort into our TITAN analysis, including fitting additional peaks and performing more steps in the bootstrap error analysis, which lowered FabG’s error significantly. We have added a section of text describing the analysis and explaining our reasoning for the increased error in some of the titrations. To reflect this, we have pointed out the additional steps of bootstrap analysis, increasing to 300 steps. We have also added the following selection of explanations. “It was observed that the FabG, FabF, and TesA resulted in greater error in the analysis. We suspect this error is due to the instability of the partner protein seen in the final titration step. FabG in particular had all of the components at extremely high concentrations in order to ensure saturation.” We have also expanded the explanations of the NMR preparation and experiments in general.

Item 1.3.A. Figures are incompletely or incorrectly referenced in the text and supplement. In the text, Figures S9, S11, S12, S13, S14, and S15 are not referenced, nor are Tables S8 and S9.

Thank you for this note. We have added reference to these supplemental figures in the main text.

Item 1.3.B. In the supplement, none of the supplemental figures are referenced in the Supplementary Methods, and of the nine supplemental tables, only Table S7 is referenced.

(line 187) “The most perturbed most residues (Table S5, Fig. S7) on AcpP were: N25 and F28 of loop 1; D35, S36, L37, T39, V40, E47, and F50 on helix 2; and T64, Q66, and A68 on helix 3.” Probably better as: ““The residues of C8-AcpP exhibiting the largest CSPs (Table S5, Fig. S2) were: N25 and F28 on loop 1; D35, S36, L37, T39, V40, E47, and F50 on helix 2; and T64, Q66, and A68 on helix 3.” [note that the figure was incorrectly cited]

We have changed the specified section of text to read, “Nearly all residues on loop II and through to helix IV are perturbed until the bottom of helix IV. The residues of C8-AcpP exhibiting the largest CSPs (Table S5, Fig. S2) were: N25 and F28 on loop 1; D35, S36, L37, T39, V40, E47, and F50 on helix 2; and T64, Q66, and A68 on helix 3.”

Item 1.3.C. (line 194) “and a one to one stoichiometry(Table S4)” correct to: “and a one to one stoichiometry (Table S9)” [incorrect table cited]
(line 211) “(Table S4)” should be “(Table S9)”

We have corrected the reference in the text.

Item 1.3.D (line 218) “Perturbations predominantly occurred in unique locations compared to FAB enzymes (Table S6, Fig. S10)” would be clearer as “Upon titration with TesA, the pattern of c8-AcpP CSPs occurred predominantly in residues different from those perturbed by FAB (Table S6, Fig S3). [note incorrect figure citation]

We have changed the text in accordance with your recommendation to read, “Upon titration with TesA, the pattern of C8-AcpP CSPs occurred predominantly in residues different from those perturbed by FAB (Table S6, Fig. S3,S10), the perturbations are relatively minor throughout with small perturbations in helix and loop I.”

Item 1.4.A. The text is poorly written in places as shown by examples. Pages in the supplement are not numbered.

We have added page numbers.

Item 1.4.B. Examples of non-sentence (line 93): “With distinct residues involved in specific classes of enzyme binding, and internal residues that are likely important to the mechanistic process of chain flipping.”

We have added to this sentence to make the subject clear, “With distinct AcpP residues involved in specific classes of enzyme binding, and internal AcpP residues that are likely important to the mechanistic process of chain flipping.”

Item 1.4.C. And (line 248) “Where AcpP and partner proteins are binding and dissociating fast enough to resolve as a single migrating peak on the spectra, rather than two distinct peaks.”

We thank the reviewer for pointing out this error in composition. We have changed the sentence to read, “. In both fast and fast-intermediate exchange, interactions between AcpP and partner proteins are occurring rapidly enough that residues resolve as a single migrating peak on the spectra, rather than two distinct peaks.”

Item 1.4.D. (line 146) “FabF displays perturbations throughout the AcpP, beginning at the end of helix I” would be more accurate as, “Upon titration with FabF, residues from C8-AcpP display chemical shift perturbations beginning at the end of helix I”

We have changed the text as suggested to read, “Upon titration with FabF, residues from C8-AcpP display chemical shift perturbations beginning at the end of helix I, with very little interaction on loop I until nearly the beginning of helix II.”

Item 1.4.E. (line 149) “the visible residues are perturbed nearly throughout the AcpP” better as “signals assigned to residues nearly throughout C8-AcpP were perturbed”

We agree that the wording should be changed. We have modified it according to the suggestion, “Helix II displays perturbations throughout until there is a small loss at the end of helix II, signals assigned to residues nearly throughout C8-AcpP were perturbed until the end of helix IV where the perturbations drop off.”

Item 1.4.F. (line 154) “(residues identified from the models will be denoted by apostrophes throughout)” would be clearer as “(residue designators for the partner proteins in the complexes will be denoted by primes)”

Thank you for this suggestion. We are happy to use this clearer way to describe the shorthand for partner residues and have changed the text as suggested to, “(residue designators for the partner proteins in the complexes will be denoted by primes)”

Item 1.4.G. (line 161) “FabB performs the same ketosynthase reaction” as what?

This was in reference to the FabF section above, but clearly this needs to be defined. We have changed the text to read, “FabB performs the same ketosynthase reaction as FabF, but performs the first unsaturated elongation step.”

Item 1.4.H. (line 163) “Overall FabB displays a slightly less broad set of CSPs than FabF” Again, signals from C8-AcpP are displaying the CSPs.

Correct. This was not the appropriate descriptor. We have changed the sentence to read, “Overall signals from the AcpP•FabB titration display a slightly less broad set of CSPs than those from the AcpP•FabF titration.”

Item 1.4.I. (line 169) “PPIs of AcpP with FabB contain a large number of predominantly acidic residues at the interface: D35, D38, and E60” Awkward sentence.

We agree that this sentence could use some retooling. We have changed the text to read, “FabB’s effect on AcpP is largest on many acidic residues at the interface, D35, D38, and E60 are all among the most perturbed,” we hope this will be a clearer wording.

Item 1.4.J. (line 172) “There is a small number of internal hydrophobic CSPs from L15 and L42” probably would be better as “A small number of hydrophobic residues, such as L15 and L42, exhibit CSPs”

As suggested, we have changed the text to read, “a small number of hydrophobic residues, such as L15 and L42, exhibit CSPs.”

Item 1.4.K. (line 197) “Perturbations of AcpP by interaction with FabI are distributed throughout the ACP.” Might be clearer as: “Upon interaction with FabI, NMR signals from residues throughout C8-AcpP exhibited CSPs.”

We have changed the text as suggested to read, “Upon interaction with FabI, NMR signals from residues throughout C8-AcpP exhibited CSPs.”

Item 1.4.L. (line 223) “Overall TesA appears to effect less residues than the other proteins tested.” Probably better as: “Overall, the titration by TesA appeared to affect signals from fewer C8-AcpP residues than the other proteins tested.”

We have changed the text as suggested to read, “Overall, the titration by TesA appeared to affect signals from fewer C8-AcpP residues than the other proteins tested.”

Item 1.4.M. (line 418) “Figures S1 to S11.” Should be “Figures S1 to S15.” We have changed this supplemental figure numbering to S15.

Item 1.4.N. (line 419) “Tables S1-S6.” Should be “Tables S1-S9”

Thank you, we have corrected the table numbers.

Item 1.4.O. (line 451) Equation: suggest removing the “x” which looks like a variable as in Supplement page 2.

This is a good observation. We have changed the symbol to a “multiplication dot.” This makes the equation more easily understood.

Item 1.4.P. Supplement page ? “who’s” should be “whose”

Corrected.

Item 1.4.Q. Supplement page ? “Van Der waal’s” should be “Van der Waals” Corrected.

Item 1.4.R. Supplement: caption to Figure S4: Remove “(Larger versions in Figure S4).” We have removed this from the caption.

Item 1.5.A. Literature is not cited in several places where it would be appropriate as shown by examples.

(line 143) C8-AcpP was titrated with increasing concentrations of the FabF ketosynthase (Fig. 2a) and compared to recently published data of C8-AcpP titrated with FabB (missing ref.).

Supplement page ?: “ICM”, “Cluspro”, “HADDOCK”, etc. should be referenced.

Figure S6 needs a reference. Are the data in BMRB?

Thank you for these notes. We should have been more thorough to cite specific examples of software in the supplemental information. We have added new citations in the revised SI. We have also submitted the data to the BMRB, as suggested earlier. As we hope for this to be a useful method that others benefit from, we will make all efforts for the data to be as reproducible and accessible as possible.

Reviewer #2

Item 2.1. Line 98: “...the residues on the PP that mediate binding cannot be determined by this process...”. Can you elaborate more on this? Does it refer to structural/energetic/dynamics contribution of residues?

This was meant to point out a shortcoming of the titration methodology, where we can gain wonderful information on labeled protein interactions. However, in the situation where unlabeled protein is titrated with labeled protein, the unlabeled protein is essentially invisible in the magnet. So although we can see the effect of its residues on the labeled carrier protein, we cannot see not which residues are interacting. To better explain this point, we have added the following sentence following the line referenced, “With titrations providing ample evidence of the carrier protein’s interacting residues, but no information of which partner residues they are interacting with.”

Item 2.2. Most of the methodological details are described in the supplementary material (e.g. docking protocol, how proteins were minimized in water, which are the “expanded calculations”). However, it took me a while to figure out that. For a broad reader, it would be helpful to include explicit referral to the supplementary material in the main text.

We agree that, proper recreation of our methods would be difficult without first consulting the methods section. To make this more explicit for the reader, we have added a call to the methods in the text with the expanded sentence, “Crystal structures of previously crosslinked AcpP-PPs were used to optimize this protocol, as described more fully in the methods section.” In order to point out the expanded protocol in our workflow figure, we have also added the line in Figure legend 3A, “The optimized docking protocol is described in more detail in the methods section.”

Item 2.3. They compared the docking model with available cross-linked structures. Which is exactly the purpose of this? Is it because the cross-linked structures are not representing the biological complex? Or is the purpose to check the docking performance? If it is already mentioned, I missed it, so it should be more clearly stated.

We began by using the crosslinked structures as a way to check the docking performance. We have attempted to make this point clearer in the language of the text. In response to this issue, we have added additional language to more explicitly state our goal with the crosslinked structures in the results, “Crosslinked structures were used in benchmarking as they give a learning set to examine which protocols and docking methods perform well, also demonstrating the ability of our docking method to appreciate ACP!PP interfaces.” We have also changed the first sentence of our discussion in an attempt to explicitly state the goal of the docking, “Our studies found that across 4 examples of recreating the interface of ACP•PP was feasible and the models were in agreement with experimental structures.” We thank the reviewer for

pointing out the unclear language and hope our edits better state the goals of the paper.

Reviewer #3

Item 3.1. There may be a catch 22 to resolve here. The reason that NMR brings new information about mechanisms, as the authors describe well, may limit applications to docking. Traditionally, NMR-assisted docking relies on the assumption that CSPs of surface residues report on molecular binding through contacts at the binding sites. Here, for AcpP, many molecular interactions are at play and NMR is sensitive to all of them. Notably, the PP and its tethered load must disengage the CP core before interacting with partner proteins. It is unclear to me how the signals of residues at the surface of AcpP may be impacted by the trajectories of the phosphopantetheine arm and its cargo as proteins engage. How are these contributions to CSPs delineated from those stemming from PPI when using constraints for docking? Is it possible that some differences in CSP fingerprints for different CP partners reflect changes in phosphopantetheine behavior rather than alternative interactions? If so, how did the authors address the contributions of phosphopantetheine when creating docking models? From the abstract and introduction, I anticipated that the manuscript would precisely address these issues as a new docking protocol is promised.

This is an excellent question which speaks to the fundamentally complicated goals of work. We intended our sections exploring the varied binding surfaces to explain the reasoning that unique perturbations derive from the protein surface. We hypothesize that the mechanism of chain flipping is driven by surface interactions to affect the substrate movements necessary. Our experimental technique can very well appreciate the binding of the AcpP. However the chain flipping must be very rapid given the time scales of the interactions, the specifics of the effects of substrate leaving the AcpP would be unseen on NMR. Furthermore, to your point it is well known that the substrate of the carrier protein will significantly effect the structure of the protein, for this reason we ensured that our protocol could appreciate changes in substrate. To better reflect this point we have made more explicit reference to the methods, which will hopefully help readers identify our treatment of the AcpP cargo. All of the docking protocols, and most others available, are not designed to have non-amino acid cargo bound to proteins being docked. However, the docking protocols were designed to appreciate the structural effects, with the protein preparation including the substrate before only deleting for the docking experiment. We have added the following text to the manuscript in order to more explicitly describe the protocol “Docking simulations were carried out between the x-ray structural model of heptanoyl-AcpP-C7 (PDB 2FAD) to which a methylene was added to the acyl chain simulate C8-AcpP and crystal structural models of binding partner proteins from which cofactors had been removed.” We have

also expanded the supplemental information to include this. This protocol can easily be modified to analyze other chain lengths in docking, with only the preparation of the carrier protein requiring modification.

Item 3.2. Please include a Method section and expand on what you currently provide in the SI. What is the buffer for modification of AcpP and the temperature and timing of reaction, what are the concentrations of reagents? etc., etc. How were NMR titrations conducted? Did you mix saturated and reference samples to create the points in between? Was the data acquired chronologically (1:0, 1:X, ..., 1:max) or scrambled in time? How was the integrity of NMR samples monitored (e.g. overlay of 1Ds at the beginning and end of titrations/acquisition or SDS-PAGE or MS before and after NMR)?

In response to this point, we have expanded the details of the titration studies in the supplemental information, focusing especially on the acquisition of the spectra and preparation of the partner proteins. In our first attempts at the experiments, we encountered the most hurdles in the preparation of partner protein samples in such a manner that they would be stable to the highly concentrated conditions necessary for the mixing of all the titration components. It is correct that the more thorough our information for performing these experiments, the more our work can be verified and expanded by others. Therefore, we have moved the methods section to the text of the manuscript, in which we have included more experimental details. Also, within the text we have included more explicit calls to the methods for additional information.

Item 3.3. As you present a new method, please provide a detailed docking protocol that simultaneously highlights the novelty of your docking procedure and that can be used by readers in a reproducible manner. The section presented in the SI most likely provides sufficient details, but it is currently provided with little effort for clarity and flow. If your paper is in part about the method, this section should not be in the SI.

We agree that the reproducibility of the protocol is important, as the protocol is essential to implementing methodology. In response to this comment, we have expanded the docking protocol in the methods and taken care that it follows logically. We have also made efforts to split the protocol into smaller sections, separating what was a large block of text into more easily followed and consulted sections. We hope this has made reproducing our method easier.

Item 3.4. It seems the authors hesitate between using cross-linked structures as

references for validating docking models and explaining that docked models differ because the methods are different. Please resolve.

Because the docked models recreate an initial interacting complex, and not a chain-flipped state like the crosslinked structures, we felt it was important to point out the differences in the systems that we were comparing. This is why we chose to present an alignment and interface-only alignment to narrow the selection examined to only those pertinent residues at the interface.

Item 3.5. References to SI material are in general wrong. SEC traces and SDS-PAGES are mentioned when describing interactions, and tables of shifts are referred to when describing Titan analysis.

We thank the reviewer for identifying this error. We have updated references to all of the SI figures and tables in the paper such that they are now correct.

Item 3.6. Tables have no captions.

We have added captions for each table to make the information presented clearer.

Item 3.7. You report a K_d of $20 \pm 40 \mu\text{M}$.

This item is similar to Item 1.2.J. above. We have performed additional steps of bootstrap analysis, finding that the FabG error was greatly improved. The new value is within the error of the previously reported K_d , although this is unsurprising given that the error was unacceptably large. We have further described the TITAN analysis in more detail in the interest of ensuring that anyone following our procedures benefits from our findings.

Item 3.8. Although the quality of figures in SI is too poor for rigorous assessment, your data appears to often display intermediate/fast exchange and not fast exchange.

We had described the exchange as fast as a generalization and hoped the TITAN off rates could be used for further judgement. However, this point is correct. We have added a discussion to the text about exchange, “the weak nature of these interactions is necessary for the “fast” or “fast-intermediate” exchange NMR chemical shifts and agrees with both our new data and previously known AcpP binding affinities. In both fast

and fast-intermediate exchange, interactions between AcpP and partner proteins are occurring rapidly enough that residues resolve as a single migrating peak on the spectra, rather than two distinct peaks. The titration effect on lineshape suggests that the interactions are not so rapid that the titrations are happening in full “fast exchange”. This is also reflected in the TITAN derived koff rates.”

Item 3.9. “Perdeuterated AcpP was used to boost signal from quenching.” Reword (not the right spectroscopy).

Excellent point. We have reworded this description to read, “Perdeuterated AcpP was used to boost signal from sensitivity lost due to the titrated partner protein. This also lowers the concentration of partner protein necessary to accommodate stability concerns.”

Item 3.10. Please relate explicitly the orientations of AcpP that are presented, in particular for surface representations (preferably in captions). I am guessing 2E has the same orientation as 2D with a reduced scale and that 4B uses the same orientations.

It is correct that the orientations in 2E, 2D and 4B are the same. We have added language to specify the orientation used.

Item 3.11. About the SI. The figures in the supporting information need to be revised for clarity; I managed to assess the take-home points but the data cannot be used and interpreted in full by a reader:

-In all spectra, increase the line thickness for the contours. Then verify that the choice of colors makes them legible. Currently, the lines are so thin that we cannot distinguish colors. Also, change the font for assignments to sans-serif and make them bold or increase the size: I had to guess quite a few.

We appreciate these comments. We have taken several steps to improve the accessibility of our supporting information. In addition to the modifications suggested, we have also deposited the titration information to the BMRB in order to make the experiments broadly accessible.

Item 3.12. -Verify that all axes labels are legible. This is not the case for CSPs nor SEC traces.

-The purity of all samples in general is well demonstrated. For FabI, Figure S4, there is a shoulder in the SEC, and the SDS-PAGE is set for sensitivity rather than resolution: if you have a diluted SDS-PAGE to show that fractions 10-13 do not contain two

overlapped bands, show it as well. Otherwise, please discuss the shoulder in SEC. TesA is probably fine given its SEC but the SDS-PAGE is psychedelic and of not much use. Please replace.

We appreciate this point. We have increased the resolution of the SI figures and adjusted the font in order to make them more legible. The reviewer raises a valid concern regarding FabI purification. Fortunately due to the high concentration of the elutions, the center of the peak at fractions 12, 13, and 14 were able to be chosen to avoid this shoulder. This was done both out of concern for purity and in order to shorten the long times required to reach the necessary concentrations for the titration experiments. Unfortunately, our gel from the day of the TesA titration became distorted when run. We have clarified in the figure legend that we were able to focus on only the cleanest bands. We have included a gel which was run on a test purification prior to the full NMR experiment. Though this one is less clean it demonstrates the strong bands which we had seen previously. Given our prior experience with the protein and clean FPLC trace we accepted the gel results prior to the titration.

Item 3.13. Please verify that the captions in SI are clear. The following sentences are unclear: Figure S11 :” Due to the experiments presented octanoyl-loaded AcpP was purified fresh for each titration.”, Figure S2 “The Y-axis is held at the 0.18 for all titrations performed.” As mentioned, all Item s should have captions. I am not sure how to interpret S7 for example.

In response to this point, we have ensured that all items in the SI have captions. We have further completed a round of examining each item for clarity. In addition, we have rewritten the caption for S11.

Item 3.14. About clarity (throughout manuscript): Some wording is misleading or confusing.

“This work reveals the molecular basis of six discrete binding events responsible for [...]” and similar statements suggest that there was no information about AcpP in complex with partners. There are structures for all systems (maybe cross-linked, maybe with holo forms, but they exist), and two NMR titrations have already been published by the authors.

Similarly, line 103 “that have eluded experimental structural characterization” must refer to a long term objective for other interactions than those presented here. The statement added to my confusion.

We may have misstated the existing structural information. We have changed the section in line 103 to read, “We sought to develop protein-protein docking protocols with Molsoft’s ICM software to predict structures of the AcpP•FabI, AcpP•FabG, and AcpP•TesA complexes that have eluded experimental structural characterization.”

On that note we have also deposited the models generated in order to make them broadly accessible. The language in the abstract implies that there was not existing information, so we have slightly changed the wording to reflect the work better, “This technique describes and compares the molecular basis of six discrete binding events responsible for *E. coli* FAB and offers insights into a method to characterize these events and those in related carrier protein-dependent pathways.”

Item 3.15. I am not sure what is combinatorial in the approach that the authors present.

We have revised the text to explicitly state that is meant by the combination of NMR spectroscopy and high resolution docking. We have changed the first sentence of the combinatorial method section to read: “To judge the ability of a combined NMR and docking method to accurately predict the structures of interacting enzymes, docking was expanded to include CSP information.”

Item 3.16. Line 268 “the ubiquity of the active sites of modular synthases is such that even when partner residues are not known the correct binding site can be inferred.” You need to develop your thoughts. This sentence could mean that active sites are conserved to the point that there is no need to identify them.

We thank the reviewer for identifying this lack of clarity. We have changed the text to read, “While our study utilized a system that has a breadth of known information, the active sites of modular synthase classes are largely conserved. Often making inferring a partner protein’s active site possible even without experimentally demonstrated residues.”

Item 3.17. Another round of proofreading is needed:

Line 51: “ [...] enzyme players (Fig. 1a). While presenting a combinatorial [...]”

line 90-91: “[...]interactions, both in carrier protein-howemediated biosynthesis [...]”

We have corrected the highlighted mistakes.

Item 3.18. I recommend working on the choice of references to more accurately assign central findings to the original papers. Currently, it appears that ACP substrate sequestration was a rather recent finding when it isn't. The choice of refs 34 and 35 to depict general applications of CSPs for PPI studies is inappropriate. Either cite again the reviews you mention (which are indeed a great choice) or cite seminal work mentioned in these reviews. Also, Markley and Co., as well as Prestegard and Co. or Kim and Co. or Yang and Co. made important contributions to ACPs mechanisms and demonstrated well the use of NMR for such studies. Citing these efforts may simultaneously restore the historical accuracy of ACP studies and provide examples of NMR applications.

In response to this comment, we have added more of the seminal work in the study of substrate sequestration and NMR studies of AcpP in general. We have also modified the mentioned section with references 34 and 35 to better focus on the subject at hand.

REVIEWERS' COMMENTS:

Reviewer #1 (Remarks to the Author):

The revised manuscript is much improved. I suggest providing the urls for BMRB and the Model Repository.

Minor suggestions:

Manuscript

Site: "actual" -- suggested

Line: 87 "2D and 3D structures of the protein" -- amino acid sequence and 3D structure of the protein

Line: 98 "however," -- however;

Line: 100 "about which partner protein" -- about the residues of the partner protein

Line: 120 "Furthermore, it should be noted that the crosslinked and apo partner proteins have differing structural similarity. -- Furthermore, it should be noted that the partner proteins in their apo (uncrosslinked) and crosslinked states exhibit structural differences.

Line 132: "binding and" -- binding, and

Line 145: "interfaces the" -- interfaces, the

Line 153 "C8-AcpP was titrated" -- 1H-15N NMR spectra were collected for C8-AcpP titrated

Line 159: "helix II, signals" -- helix II; signals

Line 161: "Residues" -- residues

Lines 170 and 222: "and an approximately one to one" -- and an approximate one-to-one

Line 548: "laboratory" -- Laboratory

Line 552: "Cells" -- cells

Line 558: "rotating at 4°C" -- with stirring at 4 °C

Lines 587, 601, 616, and 631: "filters, a Nanodrop" -- filters; a Nanodrop

Line 595: "NaN3 and" -- NaN3, and

Line 676: "hand checked a final" -- hand-checked, a final

Line 697: "TesA structures" -- TesA, structures

Line 690: "was used but AcpP" -- was used, but AcpP

Line 694ff: "Specifically: FabI and FabG were docked as tetramers. FabA, FabZ, FabF, and FabB were docked as dimers. TesA was docked as a monomer." -- Specifically: FabI and FabG were docked as tetramers; FabA, FabZ, FabF, and FabB were docked as dimers; TesA was docked as a monomer.

Line 699: "Before docking the proteins" -- Before docking, the proteins

Line 704: "After this the" -- After this, the

Line 713: "are performed" -- were performed

Line 762: "As like ICM HADDOCK" -- As with ICM, HADDOCK

Supplement

Page 3; caption to Figure S1: "CSPs greater than this value are shown in red." --- CSPs greater than this value are shown in colors as described by the inset.

Page 4 caption to Figure S2: "CSPs greater than this value are shown in red." --- CSPs greater than this value are shown in colors as described by the inset.

Page 5 caption to Figure S3: "CSPs greater than this value are shown in red." --- CSPs greater than this value are shown in colors as described by the inset.

Page 8; Figure S6: Are these assignments archived in BMRB? If so, what is the accession code?

Reviewer #2 (Remarks to the Author):

The authors have responded satisfactorily to all my concerns.

Reviewer #3 (Remarks to the Author):

The authors have addressed my concerns. Congratulations for this nice work. There are some typos:

"a methylene was added to the acyl chain simulate C8-AcpP" probably "to simulate"

No verb in "With the FabF structure 1.3Å RMSD between crosslinked39 and uncrosslinked40, FabB 3.5Å RMSD between crosslinked34 and uncrosslinked41, and FabA 4.6Å RMSD between crosslinked35 and uncrosslinked42 structures."

Figure S12-S15: increase line thickness.